Subject Area:
immunology

Keywords:
Notch, T cell, infection, autoimmunity, alloimmunity, transplantation

Author for correspondence:
Ivan Maillard
e-mail: imaillar@pennmedicine.upenn.edu

# Notch signalling in T cell homeostasis and differentiation

Joshua D. Brandstadter and Ivan Maillard

Division of Hematology-Oncology, Department of Medicine, Abramson Family Cancer Research Institute, University of Pennsylvania Perelman School of Medicine, Philadelphia, PA 19104, USA

JDB, 0000-0003-0652-5791; IM, 0000-0003-1312-6748

The evolutionarily conserved Notch signalling pathway regulates the differentiation and function of mature T lymphocytes with major context-dependent consequences in host defence, autoimmunity and alloimmunity. The emerging effects of Notch signalling in T cell responses build upon a more established role for Notch in T cell development. Here, we provide a critical review of this burgeoning literature to make sense of what has been learned so far and highlight the experimental strategies that have been most useful in gleaning physiologically relevant information. We outline the functional consequences of Notch signalling in mature T cells in addition to key specific Notch ligand–receptor interactions and down-stream molecular signalling pathways. Our goal is to help clarify future directions for this expanding body of work and the best approaches to answer important open questions.

## 1. Introduction

Notch, an evolutionarily conserved cell-to-cell signalling pathway, plays multiple functions at selected stages of innate and adaptive immune cell development, as well as in regulating mature immune cell function. Through its involvement in both developing and mature immune cells, Notch is emerging as a critical actor in host defence and immune pathology.

Notch was first discovered to influence haematopoiesis based on its oncogenic role in T cell leukaemia [1,2], a corruption of its role in T cell development [3,4]. However, an emerging role of Notch signalling in mature T cells during homeostasis and immune responses in peripheral tissues has now come into view. Work has begun to illuminate how cell-type and context-specific Notch signalling shapes T cell immune responses while also driving T cell-mediated pathologies.

Here, we will review the literature to summarize prior work and to weigh available evidence for how Notch influences mature T cells in the periphery. In particular, we will discuss the importance of *in vivo*, loss-of-function strategies that have proven most reliable as opposed to *in vitro* and gain-of-function experiments. Overall, we aim to provide a clear picture of what has been established in the field and identify larger themes for how Notch functions in mature T cells.

## 2. Overview of Notch signalling

Notch is a highly conserved cell–cell communication pathway driven by juxta-crine Notch ligand–receptor interactions (figure 1). The four mammalian heterodimeric Notch receptor paralogs (Notch1–4) interact with one of five Notch ligands in the Jagged (Jag1 and Jag2) and Delta-like (Dll1, Dll3 and Dll4) families [5,6]. Notch ligands activate Notch signalling, except Dll3 which is thought to act as a natural antagonist of the pathway [5]. A mechanical force induced by ligand–receptor interactions triggers sequential proteolytic cleavages in the Notch receptor. First, an ADAM-family metalloprotease (ADAM10) targets

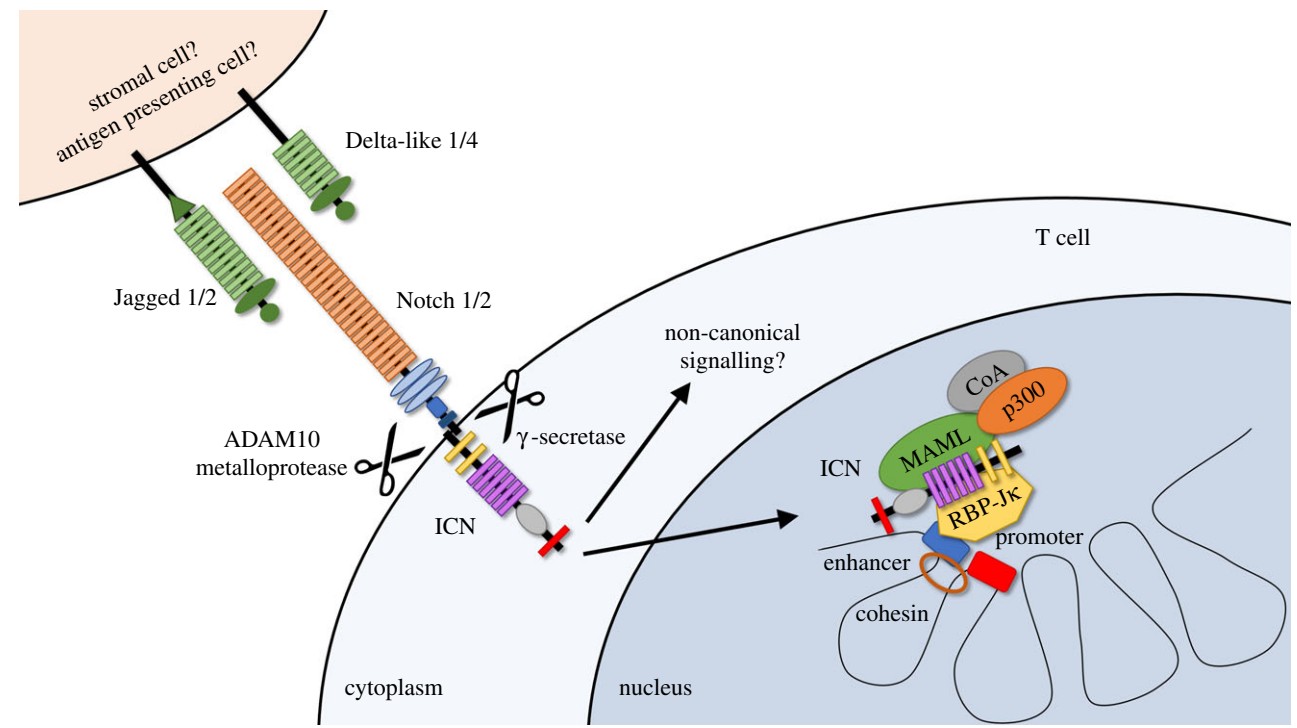

**Figure 1.** Overview of Notch signalling. Mammalian Notch receptors expressed by mature T cells receive juxtacrine signals from four activating ligands (Jagged 1/2 or Delta-like 1/4) expressed on adjacent cells (either stromal cells in secondary lymphoid organs or professional antigen-presenting cells). Ligand/receptor binding triggers sequential proteolytic cleavage of the Notch receptor, first by the ADAM10 metalloprotease and then by the γ-secretase complex. These cleavages release intracellular Notch (ICN) into the cytoplasm where it enters the nucleus to form a transcriptional activation complex with the DNA-binding transcription factor RBP-Jκ and a member of the Mastermind-like (MAML) family, which in turn recruit additional transcriptional coactivators (CoA). The Notch transcriptional complex modifies chromatin structure to form clusters of enhancers and promoters and affect transcription. In some instances, ICN was reported to signal through non-canonical RBP-Jκ/MAML-independent pathways.

the receptor's membrane-proximal extracellular domain, rendering it susceptible to the γ-secretase complex, which induces intramembrane proteolysis and releases intracellular Notch (ICN) into the cytoplasm. After migration into the nucleus, ICN interacts with the DNA-binding transcription factor RBP-Jκ and recruits a transcriptional co-activator of the Mastermind-like family (MAML1-3) [5–9]. MAML in turn interacts with other transcriptional activators, including chromatin-modifying enzymes such as histone acetyltransferases and other components of the transcriptional activation machinery.

Although transcriptional regulation by Notch signalling has been studied in multiple contexts, data from studies in Notch-driven cancers (e.g. T cell acute lymphoblastic leukaemia, B cell lymphoproliferative disorders, breast cancer) have provided the most detailed information to date. In T cell leukaemia, ICN/RBP-Jκ complexes bind thousands of sites in the genome, although less than 10% are actually dynamically regulated upon blockade of Notch signalling. Many of these dynamically regulated sites cluster with distant enhancers where Notch occupancy is associated with alterations in chromatin regulation [10]. Interestingly, recent work illuminated how oncogenic Notch can influence chromatin looping to reposition enhancers into '3D cliques' of interacting enhancer/promoter spatial clusters (figure 1) [11]. This pattern of activity broadens the mechanisms of Notch-mediated control of gene expression beyond its effects on a static cohort of target genes, suggesting that context from other signals might be important to determine patterns of enhancer activation and chromatin repositioning. Thus, individual Notch target genes are predicted to be highly context-dependent.

Notch signalling is regulated by strict temporal and spatial control of Notch ligand expression by selected cells. For example, high levels of Dll4 ligands are expressed in thymic epithelial cells, creating an anatomical niche for Notch signalling in T cell development [12–14]. Notch signals are also regulated by O-glycosylation of serine or threonine residues in the epidermal growth factor (EGF) domains of the receptor. Loss of O-glycosylation phenocopies loss of Notch signalling [15]. O-glycosylation can be elongated by the addition of N-acetylglucosamine by the glycosyltransferase Fringe, which biases Notch receptors to preferentially signal via Delta-like over Jagged ligands [16]. Genetic deletion of *Fringe* genes typically induces Notch loss-of-function phenotypes, including effects on T cell development [17].

After initial proteolytic activation, Notch signalling is regulated by the rapid targeting of active ICN for proteasomal degradation through its C-terminal PEST domain via the FBW7 E3 ubiquitin ligase. *FBW7* mutations and truncations of the *NOTCH1* PEST domain have been identified in Notch-driven T cell acute lymphoblastic leukaemia (T-ALL). Over 50% of all T-ALL patient samples and cell lines carry activating *NOTCH1* mutations, including PEST truncations and membrane-proximal mutations that induce receptor activation [2,18,19].

Notch signalling is essential for T cell development, and its effects can be corrupted to drive T-ALL [2]. In this context, which is not the central focus of our review, key mutational mechanisms of oncogenic Notch activation include point mutations affecting the NOTCH1 extracellular heterodimerization domain, thus leading to constitutive proteolytic receptor activation; and mutations truncating the intracellular

C-terminal NOTCH1 PEST domain, increasing its half-life after activation. These two classes of mutation can occur together in the same *NOTCH1* allele, suggesting cooperativity. Altogether, activating *NOTCH1* mutations and related genetic events have been reported in at least 60–70% of primary T-ALL cases, consistent with a dominant oncogenic function of Notch signalling in this disease.

Similarly, *NOTCH1* and *NOTCH2* mutations are found in different B cell lymphomas, with *NOTCH1* being recurrently mutated in chronic lymphocytic leukaemia (CLL) and *NOTCH2* in marginal zone lymphoma [20–25]. These mutations are typically frameshift or nonsense mutations decreasing ICN turnover by truncating the C-terminal PEST domain to make ICN less vulnerable to degradation. Interestingly, even independent of explicit *NOTCH1* mutations, roughly 50% of CLL cases in one series had a *NOTCH*$^{high}$ gene expression signature and another series found increased activated NOTCH1 by immunohistochemical staining in over 80% of cases, suggesting that Notch ligands drive signalling in CLL even with a non-mutated Notch receptor [26,27].

## 3. T cell development

T cell development proceeds after lymphocyte progenitors differentiate from bone marrow haematopoietic stem cells and migrate to the thymus [28]. Specialized thymic epithelial cells induce T cells to develop along an organized stepwise approach. The process begins with early thymocytes that are double-negative for cell surface expression of CD4 and CD8. The cumulative effect of this process is the generation of CD4$^+$ and CD8$^+$ T cells with a diverse repertoire capable of recognizing peptide–MHC antigen complexes.

Notch has been traditionally described for its role in early T cell development. This topic has been reviewed elsewhere [29], but we will highlight critical observations relevant for the interpretation of insights in mature T cells. Notch signalling plays a critical role during T lineage commitment in the thymus, a role that was first thought to be at the expense of B lymphopoiesis. Genetic inactivation of *Notch1* or its downstream transcriptional machinery results in a hypoplastic thymus permissive for intrathymic B lineage development [3,30]. Reciprocally, overexpression of constitutively active Notch results in the development of thymic-independent T cells and suppression of bone marrow B cell development. However, Notch also exerts negative regulation of myeloid fates in the thymus [31,32]. Thus, the original model of Notch controlling a T/B binary cell fate decision gave rise to a more complex pattern of Notch driving T lineage development while repressing multiple alternative cell fates. Interestingly, recent work reported a role for Notch signalling during initial prethymic steps of lymphoid and T lineage specification in the bone marrow, suggesting that endothelial and non-endothelial mesenchymal elements in the bone marrow microenvironment can provide signals to lymphoid progenitors through the Dll4 Notch ligand [33,34]. Yet, the abundance of Dll4 expression in the bone marrow and/or its precise microanatomical distribution must remain tightly controlled, as broad Dll4 derepression in LRF-deficient mice can lead to full-fledged extrathymic T cell development at the expense of other lineages [35,36].

As lymphoid progenitors enter the thymus, they encounter dense expression of Notch ligands on cortical thymic epithelium that is essential for thymopoiesis [3,13,14,37,38]. Notch signals persist until the pre-T-cell receptor checkpoint, after which Notch signalling intensity decreases [39–42]. Double-positive thymocytes receive little Notch signalling under physiologic conditions. Notch is dispensable for positive and negative selection [43–44] and for release of mature T cells into the periphery.

## 4. Notch signalling impacts mature T cell function

An important role for Notch signalling in mature T cells has now been established in models of host defence, autoimmunity and alloimmunity (table 1). Work describing how Notch functionally impacts mature T cells began with overexpression experiments, leading to conclusions that were then revised based on *in vivo*, loss-of-function strategies.

Early evidence for a role for Notch in mature T cells suggested a tolerogenic effect. Overexpression of Jagged1 in professional antigen-presenting cells (APCs) expressing a host dust mite antigen conferred tolerance and drove CD4$^+$ T cell differentiation into immunosuppressive regulatory T cells (Treg) [72]. Similar tolerance and Treg differentiation were observed when Jagged1-overexpressing APCs presented Epstein–Barr virus antigens to autologous or allogeneic T cells in culture, suggesting a role for Notch in tolerance to virus and alloantigens [73,74]. However, these overexpression models did not correlate well with other *in vitro* findings that Notch ligands in APCs were upregulated in response to pathogen encounter, suggesting proinflammatory functions [75–77].

More recently, genetic and pharmacologic *in vivo* loss-of-function strategies have shown a dominant proinflammatory effect of Notch signalling in mature T cells. Notch signalling appears important for a robust T cell response in host defence (table 1). For example, conditional deletion of *Notch1* and *Notch2* (but not either in isolation) in mouse CD4$^+$ T cells conferred susceptibility to *Leishmania* infection [47], suggesting that Notch is essential to generate IFNγ-secreting cells to control the parasitic infection. Interestingly, the *Dll1* gene has been linked to susceptibility to visceral leishmaniasis [78]. In other infection models, *Notch2* loss impaired CD8$^+$ T cell function and conferred susceptibility to the parasite *Trypanosoma cruzi* [45]. Genetic *Notch1* and *Notch2* inactivation impaired CD8$^+$ T cell response to both *Listeria* and influenza infection, ultimately impairing pathogen clearance [48,50]. *In vivo* administration of a neutralizing antibody against Dll1 also impaired clearance of influenza infection and resulted in higher mortality and lower production of IFNγ, a finding replicated upon *in vivo* pan-Notch inhibition via γ-secretase inhibitors (GSI) [49]. Expression of a dominant-negative form of MAML (DNMAML) in CD4$^+$ T cells to block Notch signalling resulted in higher fungal burdens from *Cryptococcus neoformans* [51]. Similarly, a neutralizing antibody against Dll4 impaired immune control of mycobacteria-elicited pulmonary granulomatosis with lower levels of otherwise protective cytokines found in these mice [46]. Impairment in the CD4$^+$ T cell response was also seen with model antigens upon conditional deletion of *Notch1/2* or *Rbpj* (versus deletion of *Dll4* in APCs) [70,71]. Altogether, loss-of-function approaches have established proinflammatory effects of Notch signalling on mature T cells in host defence, but with context-specific consequences.

royalsocietypublishing.org/journal/rsob   *Open Biol.* **9**: 190187

**Table 1.** Experimental evidence supporting a role for Notch signalling in mature T cell function.

| | disease model | *in vivo*, loss-of-function strategy | outcome | citation |
|---|---|---|---|---|
| host defence | *Trypanosoma cruzi* infection | deletion of *Notch2* in CD4$^+$ T cells | impaired survival; impaired GZMB expression | [45] |
| | mycobacteria-elicited pulmonary granulomatosis | systemic anti-Dll4 treatment | larger granulomas; decreased Th17 cytokines (IL-17/17A/F, -6, -21) | [46] |
| | *Leishmania major* infection | deletion of *Notch1* and *Notch2* in T cells | susceptibility to infection, impaired IFNγ | [47] |
| | *Influenza* infection | deletion of *Notch1* and *Notch2* in CD8$^+$ T cells; systemic anti-Dll1 treatment; GSI treatment | impaired viral clearance; decreased terminal effector cell differentiation, impaired IFNγ; impaired survival | [48,49] |
| | *Listeria monocytogenes* infection | deletion of *Notch1* and *Notch2* in CD8$^+$ T cells | decreased short-term effector cell differentiation, impaired IFNγ | [50] |
| | *Cryptococcus neoformans* infection | DNMAML expression in T cells | increased fungal burden, impaired Th1/Th2 response | [51] |
| autoimmunity | experimental autoimmune encephalomyelitis (multiple sclerosis) | GSI treatment; systemic anti-Dll4 treatment; systemic anti-Notch3 treatment; DNMAML expression in T cells | decreased disease scores, impaired IFNγ; impaired IL-17A/IFNγ in CNS | [52–57] |
| alloimmunity | graft-versus-host disease | DNMAML expression in donor T cells; deletion of *Notch1* and *Notch2* or *Rbpj* in donor T cells; systemic anti-Notch1/2 or anti-Dll1/4 treatment; deletion of *Dll1/4* in recipient CCL19$^+$ stromal cells | decreased GVHD severity scores, increased survival, impaired IFNγ, increased Treg expansion | [58–66] |
| | heart transplant | systemic anti-Dll1 treatment + CTLA4-Ig/CD28KO; systemic anti-Dll1/4 treatment; DNMAML expression in T cells; systemic anti-Notch1 ± CTLA4-Ig treatment; Deletion of *Notch1* in Tregs | delayed cardiac allograft rejection, impaired GZMB; impaired IFNγ/IL-4, decreased graft infiltration, decreased donor-specific alloantibodies | [67–69] |
| | lung transplant | systemic anti-Notch1 + anti-CLTA4-Ig treatment | preserved airway patency with less lymphocytic infiltrations and delayed lung allograft rejection | [69] |
| | human skin graft to mouse with chimeric human haematopoietic system | systemic anti-Notch1 | decreased T cell infiltration, greater proportion of Tregs infiltrating, improved vascularity | [69] |
| model antigens | H-Y antigen | deletion of *Dll4* in DCs | impaired CD4$^+$ T cell activation and IL-2 production | [70] |
| | keyhole limpet Haemocyanin alum, *Schistosoma mansoni* egg extract | deletion of *Notch1* and *Notch2* or *Rbpj* in T cells | impaired CD4$^+$ T cell activation | [71] |

*In vivo* loss-of-function experiments in autoimmunity models also revealed a dominant proinflammatory role of Notch signalling in mature T cells (table 1). In a mouse model of multiple sclerosis, Notch inhibition by γ-secretase inhibitors (GSI) slowed disease progression [52]. This exciting finding was supported by later studies suggesting roles for Dll4 and Notch3 via systemic blocking antibodies [52–56]. We also observed dramatic disease protection in mice with T cells deprived of Notch signalling via expression of DNMAML [57]. This protection appeared independent of T cell activation and differentiation in secondary lymphoid organs, although Notch-deprived, myelin-reactive T cells in the central nervous system did not produce inflammatory IL-17A and IFNγ, suggesting local effects of the Notch pathway in the target

organ. Clinical correlations and other associative studies also suggest roles for Notch in other autoimmune disorders, including rheumatoid arthritis and systemic sclerosis, and indeed GSI treatment allowed for disease improvements in mouse models [79–81]. However, these autoimmune disease models require further work using more targeted loss-of-function strategies to clarify the role of Notch signalling in this context.

Notch signalling in T cells is a critical regulator of alloimmunity during both solid organ transplant rejection and graft-versus-host disease (GVHD) after allogeneic bone marrow transplantation, which we have reviewed elsewhere [82]. As in models of host defence, early research relied on overexpression of Notch ligands, which suggested a tolerogenic effect of Notch signalling in T cells. Fibroblasts overexpressing both Dll1 and MHC-loaded alloantigens could be adoptively transferred to mice with cardiac allografts to delay CD8+-mediated graft rejection [83]. Furthermore, Jagged1-overexpressing APCs (alongside CD40 blockade) delayed cardiac allograft rejection [84]. However, as observed in infection models and autoimmunity, *in vivo* loss-of-function approaches instead revealed a dominant proinflammatory role of Notch signalling in alloimmunity (table 1). The use of Dll1-blocking antibodies alongside costimulation blockade delayed rejection and reduced inflammatory cytokine secretion in a MHC-mismatched heart transplantation mouse model [67]. Similarly, more complete Notch blockade with either Dll1 and Dll4 blocking antibodies or T cell-specific ablation of Notch signalling using DNMAML delayed cardiac allograft rejection without costimulation blockade [68]. Interestingly, short-term Dll1/4 blockade during the peritransplant period was sufficient to confer projection against CD4+ T cell-mediated rejection in this model. Similarly, recent work showed a role for peri-transplant Notch1 antibody blockade in delaying graft rejection in models of MHC-mismatched heart and lung transplantation in addition to a model where human skin is grafted onto a mouse with a chimeric human haematopoietic system [69]. This work found that Treg depletion through anti-CD25 antibodies cancelled the beneficial effects of Notch blockade and that conditional *Notch1* deletion in Tregs could similarly delay graft rejection.

GVHD is the life-threatening consequence of an alloimmune response following bone marrow transplantation, whereby donor T cells attack recipient tissues [82]. We found dramatic protection against GVHD in MHC-mismatched transplants when Notch signalling was blocked in donor T cells via either conditional DNMAML expression or loss of RBP-Jκ [58,59]. We phenocopied these effects via conditional *Notch1* and *Notch2* deletion in donor T cells [60,61]. We also obtained similar protection using blocking antibodies against Dll1 and Dll4 ligands. A dominant role was observed for Notch1 and Dll4. Interestingly, transient Dll1/4 blockade at the time of transplant conferred long-term protection from GVHD. Other groups identified protection from GVHD upon abrogation of Notch signalling in mature T cells in their models [62–64]. While one model used conditional loss of RBP-Jκ in Tregs to suggest that Notch signalling in Tregs is the essential driver of GVHD protection [64], side-by-side conditional deletion of Notch signalling in Tregs alongside Tconvs revealed that Notch inhibition in Tconvs remained essential for conferring protection from GVHD [65].

Overall, across models of infection, autoimmunity and alloimmunity, *in vivo* loss-of-function experiments revealed a pleomorphic proinflammatory function of Notch signalling

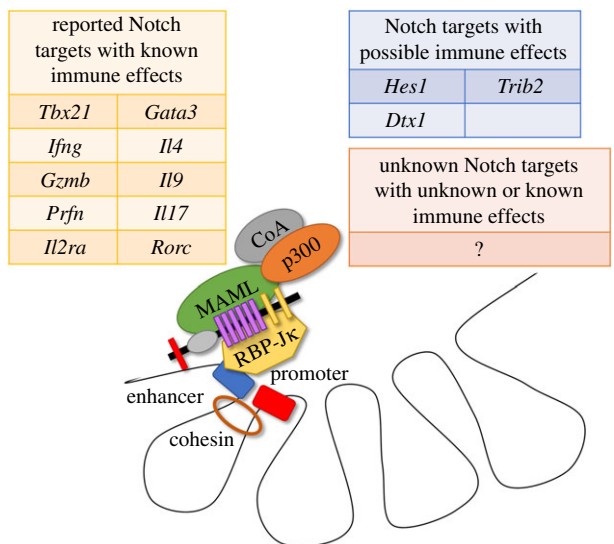

**Figure 2.** Classes of possible Notch transcriptional target genes in mature T cells. Notch dynamically regulates transcription by binding RBP-Jκ and a member of the MAML family to recruit additional coactivators and globally alter chromatin structure at clusters of enhancers and promoters. Most genome-wide research into Notch transcriptional targets has been performed in developing thymocytes and Notch-driven T cell leukaemia, with unclear significance for mature T cells. In this context, Notch has been shown to alter transcription of selected immunologically important genes related to T cell differentiation and function. Additionally, a small number of classically described Notch targets have purported immunological effects in mature T cells (e.g. *Hes1*, *Dtx1*, *Trib2*). However, many immunologically relevant Notch transcriptional targets probably remain to be defined in mature T cells.

in mature T cells. Collectively, this work shows the power of *in vivo* loss-of-function models to uncover physiologic roles for Notch signalling as overexpression, gain-of-function experiments had suggested an otherwise contradictory tolerogenic role for Notch in these settings.

# 5. Notch intracellular signalling pathways in mature T cells

Mature naive CD4+ and CD8+ T cells express Notch1 and Notch2 receptors [77,85–87], with upregulated expression following T cell receptor (TCR) stimulation [88]. Increased cleaved Notch1 was also observed following antigenic stimulation of CD4+ T cells [85]. Besides Notch1 and Notch2, Notch3 has been explored for its effects in developing thymocytes [89] and in T-ALL [90], and at least one group has also detected it in cultured mature T cells [88]. However, while systemic Notch3 blocking antibodies was protective in a mouse model of multiple sclerosis [55], no genetic loss-of-function data have been reported to date demonstrating its role in mature T cells. In contrast, forced Notch3 overexpression experiments have suggested a potential role that needs to be further explored [91,92].

In mature T cells, as in developing thymocytes, Notch signals via proteolytic cleavage of the receptor to release ICN, which translocates to the nucleus and initiates transcriptional activation of target genes. The nature and regulation of Notch transcriptional targets remains poorly understood in mature T cells (figure 2). Gain-of-function experiments have shown increased *Hes1* and *Dtx1* expression upon Notch ligand

exposure [73,93], and we have observed alterations in *Dtx1*, *Hes1*, *Il2ra* and *Trib2* gene expression using an *in vivo* model of alloimmunity with Notch ligand blockade—all genes previously identified as Notch targets in other contexts [66]. Others found ICN capable of binding the *Gzmb* promoter [45]. More recently, using bulk RNA sequencing, we found that Notch inhibition during allogeneic bone marrow transplant impaired transcription of a subset of Myc-regulated genes and altered a transcriptional programme largely independent of targets identified in T cell leukaemia and thymopoiesis [65].

Notch has been shown to regulate a number of genes necessary for guiding different T cell differentiation fates, as discussed further below. Indeed, Notch has been shown to target *Tbx21*, *Il4*, *Gata3-1a*, *Il17a* and *Rorc* (figure 2) [52,77,94,95]. In addition, Notch has been shown to directly regulate *Ifng* via binding to the *Ifng* CNS-11 enhancer [96]. However, transcriptional targets of Notch signalling remain much better studied in developing thymocytes and malignant cells. It is unclear to what extent these findings can be applied to mature T cells as many targets have been found to be cell-type and context-dependent. For example, *IL7RA* has been a well-validated Notch target gene in thymocyte progenitors and T-ALL, although it remains to be explored in mature T cells [10,97–99]. Regardless, it seems likely that, as shown in T cell leukaemia, only a small fraction of ICN/RBP-Jκ binding sites is dynamically regulated [10].

Emerging data also suggest a potential role for non-canonical Notch signalling whereby Notch exerts biologic functions independently of ICN translocation to the nucleus to drive transcription with RBP-Jκ, MAML and other coactivators [52,93,100–102]. Fruit flies and cultured mammalian cells were first reported to respond to Notch signals independent of its transcriptional activation complex [103,104]. Roles for non-canonical Notch signalling pathways have also been proposed in mature T cells [52,93,101,102]. These pathways include signalling via Tbx21 upregulation and NF-κB activation. Indeed, ICN was found to complex on the *Tbx21* promoter and interact directly with NF-κB pathway intermediates, supporting a role for non-canonical Notch signalling in mature T cells.

Overall, the most rigorous strategy to identify non-canonical Notch signalling is to show phenotypic discordance between genetic or pharmacologic abrogation of Notch receptors and deletion or impairment of its transcriptional machinery (either via loss of RBP-Jκ or DNMAML expression). For example, a role for non-canonical Notch signalling was suggested in controlling IFNγ production by CD4[+] T cells during *Leishmania* infection [47]. *Notch1* and *Notch2* inactivation in T cells conferred susceptibility to infection, but mice with a similar conditional loss of RBP-Jκ remained protected. These data provide evidence for a non-canonical pathway of Notch signalling in this setting, although precise molecular mechanisms remain to be identified. A similarly rigorous genetic approach found a limited role for non-canonical Notch signalling in Tregs, although canonical signalling remained the dominant pathway [64]. After finding that Notch1-deficient and RBP-Jκ-deficient Tregs had enhanced expansion and tolerance induction, the authors forced ICN expression in Tregs to drive Treg dysfunction and autoimmunity. Loss of RBP-Jκ in ICN-expressing cells rescued many of these phenotypes (favouring a canonical pathway). However, selected findings were unaffected, such as impaired demethylation of

*Foxp3*, suggesting a role for non-canonical Notch signalling. Overall, although non-canonical signalling occurs in some instances in mature T cells, confirmatory *in vivo*, loss-of-function studies remain essential to determine the relevance of this phenomenon in most contexts.

# 6. Notch signalling in T cell differentiation

The use of loss-of-function, *in vivo* modelling also provided clarity for how Notch functions in mature T cell differentiation. Differentiation is a polarization of the immune response toward a particular set of functions most effective at combating a particular type of threat. During differentiation, these CD4[+] T cells respond to APCs via juxtacrine cell–cell signals and soluble cytokines. Two of the most prominent differentiation states include the T helper 1 (Th1) cell fate where CD4[+] T cells, driven by the transcription factor T-bet (encoded by *Tbx21*), make IFNγ to combat viruses and intracellular pathogens [105] and the Th2 cell fate where T cells combat helminth parasites via production of IL-4, IL-5 and IL-13 through the GATA3 transcription factor [106–108]. Although others have reviewed the role of Notch in T cell differentiation, we will focus on critical evaluation of the literature to weigh seemingly contradictory evidence and draw out larger emerging themes [109].

An early model for how Notch might be influencing mature T cell differentiation suggested that Notch was a bipotential switch, toggling between Th1 and Th2 cell fates, with consequences dependent on the nature of the Notch ligands involved. Indeed, early work found that forced overexpression of Notch ligands Dll4 and Jagged1 in APCs resulted in ligand-specific differentiation of mature CD4[+] T cells to Th1 or Th2 cell fates, respectively [77].

However, a more recently proposed model whereby Notch acts as an unbiased amplifier of T cell differentiation may best correspond with available data [96]. The amplifier model helps synthesize otherwise conflicting findings that Notch controls gene expression of opposing transcriptional differentiation programmes, including playing a role in targeting *Tbx21*, *Il4*, *Gata3-1a*, *Il17a* and *Rorc* [52,77,94,95]. An amplifier model clarifies seemingly contradictory data that suggested roles for Notch signalling in promoting either Th1 or Th2 responses under different polarizing conditions. For example, Notch was reported to promote Th1 differentiation as Dll1 could drive Th1 commitment ex vivo [92], while Notch blockade via GSI suppressed Th1 differentiation *in vivo* [52]. Others proposed that Notch promoted Th2 differentiation, as DNMAML expression led to impaired IL-4 and Th2 cytokine production and impaired defence against *Trichuris muris* [94,110]. An amplifier model, whereby Notch helps sustain expression of different gene targets for both Th1 and Th2, synthesizes these data and makes sense of how a fundamental cell signalling pathway can drive differentiation of several distinct lineages. In other words, Notch regulates distinct transcriptional programmes for differentiation to various T cell fates by sensitizing T cells to cytokine-derived and other regulatory signals. This model was proposed after experiments using conditional expression of DNMAML to ablate Notch signalling found that Notch had no impact on Th2 initiation following infection with the helminth *T. muris*, but did affect maintenance of Th1 and Th2 programmes [96]. Notch was also shown to be needed for the maintenance of the Th17 response (another

royalsocietypublishing.org/journal/rsob    Open Biol. 9: 190187

royalsocietypublishing.org/journal/rsob    Open Biol. **9**: 190187

differentiation state characterized by IL-17 expression that is important in immunity against extracellular pathogens and associated with autoimmune conditions). The amplifier model is further supported by mechanistic work showing that Dll4 expression on APCs could activate CD4$^+$ T cells via augmenting PI3 K pathway signalling downstream of CD28 co-stimulation through a pattern of activity reminiscent of costimulatory signals [70].

Beyond Th1, Th2 and Th17, Notch has also been linked to other differentiation states. Th9 responses, so called for the production of the cytokine IL-9, are related to Th2 in anti-helminth immunity and appear to be implicated in certain autoimmune diseases. It also requires Notch signals as loss of Notch1/2 receptors abrogated the development of Th9 cells, while ICN, RBP-Jκ and Smad3 (downstream of TGFβ) were all found to cooperatively bind the *Il9* promoter [53].

Notch also regulates T cell differentiation into follicular T helper cells (Tfh). Tfh cells specialize in helping B cells during isotype class switching and affinity maturation via CD40 L and secreted cytokines [111]. *Notch1/2* deletion in T cells resulted in decreased numbers of Tfh and IL-21 production in response to parasitic infection and hapten immunization [112], translating into impaired germinal centre formation and IgG1 production independently of Notch's effects on IL-4.

Notch signalling may also regulate Treg differentiation, which mediates peripheral tolerance via FoxP3-dependent mechanisms [113]. Overexpression studies first suggested that Jagged ligands were capable of promoting Treg expansion [73]. Indeed, cultured T cells exposed to TGFβ, which assume a Treg-like phenotype, lost this ability upon addition of GSI [114]. However, *in vivo* loss-of-function approaches suggested an alternative role for Notch in Tregs. Genetic *Notch1* or *Rbpj* inactivation instead led to a 'super-regulatory' phenotype [64]. Consistent with this work, blocking Notch signalling *in vivo* through genetic or pharmacologic means during allogeneic bone marrow transplantation resulted in Treg expansion [58,66]. Overall, *in vivo* work suggests that Notch signalling curtails Treg function.

# 7. Source and specificity of Notch ligands

*In vivo* T cell conditional abrogation of Notch receptors and of the Notch transcriptional machinery (either via RBP-Jκ loss or DNMAML expression) has been the most robust approach to glean insights into the pleotropic effects of Notch in mature T cells. Ligand-targeted genetic strategies have proven initially more challenging, given limited insight into cellular sources of ligands and possibly different or redundant effects from different ligands. Loss-of-function strategies targeting ligands therefore mostly relied on systemic blocking antibodies. Otherwise, many experiments have depended upon ligand over-expression models, which are hypothesis-generating but less physiologically relevant.

In development, the thymic epithelial cell niche where Notch ligands are provided to early thymocytes has been well characterized (figure 3) [12–14]. The corollary niche for mature T cells in the periphery is only beginning to emerge. Initial focus centred on professional APCs as the suspected cellular source of Notch ligands, as they provide antigen and multiple costimulatory signals during T cell activation. Indeed, Notch ligands are expressed by APCs and *in vitro* experiments where Notch ligands were induced by Toll-like

receptor (TLR) agonism suggested that APCs could be the cellular source of ligands [77,115,116]. Furthermore, Notch ligands upregulated in a subpopulation of APCs from mice receiving allogeneic bone marrow transplantation could activate T cells *in vitro* in a ligand-dependent fashion [63].

However, the physiologic relevance of *in vitro* experiments suggesting APCs as the source of Notch ligands for mature T cells had not been rigorously confirmed with *in vivo* experimentation. On the contrary, loss-of-function *in vivo* experiments uncovered a non-haematopoietic source of Notch ligands. This was first suggested for marginal zone B (MZB) cells—another lymphocyte population—as bone marrow chimeras identified the source of Dll1 necessary to generate MZBs to be non-haematopoietic [117]. This was followed by an elegant series of experiments whereby *Dll1* and *Dll4* were conditionally inactivated in non-haematopoietic secondary lymphoid organ (SLO) stromal cell populations that expressed a *Ccl19*-driven Cre recombinase [118]. By lineage tracing cells expressing the *Ccl19-Cre* transgene, candidate stromal cell subpopulations were defined among Dll1-expressing cells in the spleen (CD45$^-$CD31$^-$PDPN$^-$) responsible for MZB generation and among Dll4-expressing cells in lymph nodes (CD45$^-$CD31$^-$CD35$^{+/-}$PDPN$^+$) necessary for Tfh differentiation. The role for these non-haematopoietic stromal cells in SLOs was particularly intriguing given that the only other known niche for Notch signalling in the thymus similarly depends upon Notch ligand expression by non-haematopoietic cell types (thymic epithelial cells). Past work had also found high levels of Dll1 and Dll4 expression on blood and lymphatic endothelial cell populations [117,119–122]. Endothelial cell populations had been suggested to be the source of Notch ligands in developing liver and neural stem cells [123,124]. Though endothelial cells were also implicated as the ligand source for immune cells [117], more recent work instead points to a non-haematopoietic, non-endothelial cell source [118].

In alloimmunity and GVHD, we found the cellular source of Notch ligands to be non-haematopoietic as well [66]. This finding was unexpected given prior work suggesting that donor and host APCs are critical to activate alloreactive donor T cells in GVHD [125,126]. Using *Ccl19-Cre*-driven *Dll1* and *Dll4* inactivation, we identified a non-haematopoietic SLO stromal cell subpopulation (CD45$^-$CD31$^-$PDPN$^+$CD157$^+$) as the likely cellular source of Dll1/Dll4 ligand in GVHD [66]. These fibroblastic stromal cells are believed to reside predominantly in the T zone of the SLO. It is currently unknown why these non-haematopoietic stromal cells are the physiologically relevant source of Notch ligands in allogeneic transplant. It is interesting to speculate that the expression of T-cell chemoattractants CCL19 and CCL21 may render these cells analogous to thymic epithelial cells, which express Dll4 as well as CCL21/25 and CXCL12 under the control of the Foxn1 transcription factor (figure 3) [12]. SLO fibroblastic stromal cells also secrete IL-7, which is necessary for mature T cell survival [127]. Future work should investigate how broadly relevant SLO stromal cells are as a source of Notch ligands in other immunological contexts.

A number of reports have suggested different functional effects of the four agonistic Notch ligands. Data from conditional genetic deletion are limited given uncertainty about the relevant cellular source of ligands. However, work in stromal cells found *Dll1* expression in a splenic stromal cell subpopulation critical for MZB cells, while identifying *Dll4* expression in lymph nodes as essential for Tfh differentiation

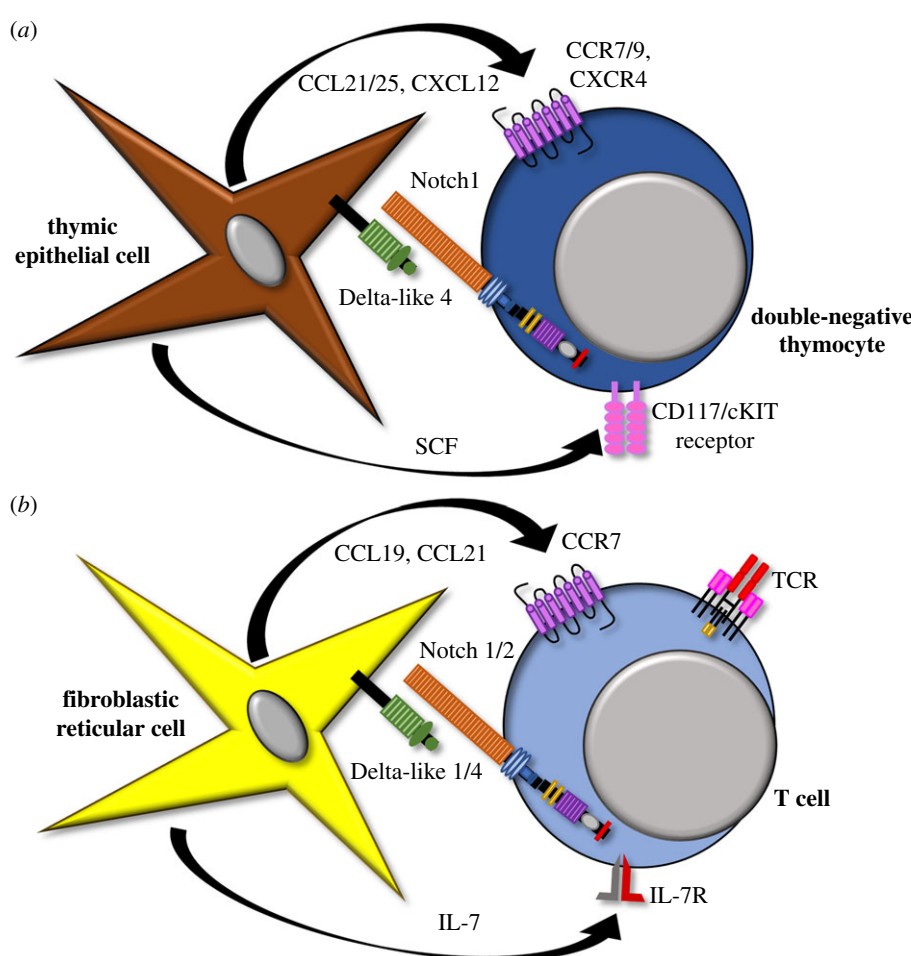

**Figure 3.** The Notch niche in primary and secondary lymphoid organs. (*a*) Developing CD4⁻CD8⁻ ('double-negative') thymocytes receive essential Notch signals from Delta-like 4 Notch ligands expressed by thymic epithelial cells. These ligands interact with Notch1 receptors in developing T cells. The specialized thymic niche for T cell development also releases chemokines to attract early T lineage progenitors and Stem Cell Factor (SCF) to support their survival. (*b*) Mature T cells receive Notch signals from Delta-like 1 and 4 Notch ligands expressed by non-haematopoietic fibroblastic stromal cells in secondary lymphoid organs. These ligands interact with Notch1 and Notch2 expressed by mature T cells. This niche also provides chemokines to attract circulating T cells in the periphery and an IL-7 pro-survival signal.

[118]. These data fit with our work that found a dominant role for Dll4 over Dll1 in alloreactivity using both blocking antibodies and selective deletion in *Ccl19*-expressing stromal cells [60,66]. Other experiments, relying upon *in vitro* and overexpression models, suggested that different ligands induce distinct patterns of differentiation, with Delta-like family members associated with a Th1 response and Jagged with Th2. However, much of this work could be confounded by TLR agonism, which selectively upregulates Delta-like family members over Jagged, making it unclear how much of the Th1/Th2 differentiation bias was instead due to independent effects of TLR agonism [76,77,128]. Blocking antibodies have also suggested different effects from different ligands. Antibodies blocking Dll1 and Dll4—and not Jagged1—could suppress deleterious effects in a multiple sclerosis mouse model [54,129]. The mechanistic basis for qualitatively different signals from Delta-like and Jagged family members, which both signal similarly through the Notch receptor, is unclear. Indeed, another group using ligand overexpression in APCs did not observe a Delta-like/Jagged bias in Th1/Th2 instruction in the absence of polarizing cytokines [130]. It may be that the seemingly different effects of the different ligands are instead attributable to different cellular sources of ligands. Overall, loss-of-function *in vivo* approaches promise to provide the most reliable and physiologically relevant answers.

## 8. Concluding remarks

A picture of how Notch contributes to the functions of mature T cells in the periphery is slowly coming into view, although many outstanding questions remain. Importantly, lessons learned from studying the role of Notch in T cell development bore significantly on the study of Notch in mature T cells and will play an important role in future experiments to resolve open questions. Specifically, the value of *in vivo*, loss-of-function experiments in this context cannot be overstated. This was seen during experiments deciphering downstream Notch signalling in T cells, mapping the role of Notch in T cell differentiation and identifying the cellular source of Notch ligands. Furthermore, the concept of Notch as a bimodal switch responsible for binary cell fate decisions had to be revised, both in developing and in mature T cells, to make way for a more nuanced picture of Notch as an amplifier of other activating signals. Finally, recent work suggested that a wider range of candidate cellular sources of ligands should be considered, as non-haematopoietic SLO stromal cells have been convincingly shown to play a role in at least in some settings, even though the bulk of the literature had focused on haematopoietic APCs.

To date, it remains unclear how broadly important stromal cells are as cellular sources of Notch ligands and to what extent

other cells, including haematopoietic APCs, provide Notch signals to T cells in the periphery. Beyond TLR signalling, it is unknown which signalling pathways or transcriptional machinery control the expression of Notch ligands to impact mature T cell function. Characterization of the niche for Notch ligand presentation, as has been done in the thymus for developing T cells, remains to be done. Additionally, the downstream targets of Notch signalling in mature T cells remain ill-defined and insights into how Notch effects gene expression often rely on extrapolation from work in T cell leukaemia and developing thymocytes. Finally, if different Notch ligands do indeed physiologically provoke different functional consequences in mature T cells, the mechanistic basis for this observation remains poorly understood. The answers to these remaining questions will become particularly relevant in revealing new ways to interfere with the Notch pathway so that it might be targeted therapeutically.

Data accessibility. This article has no additional data.

Authors' contributions. J.D.B and I.M. both contributed to the manuscript writing, revision and figure/table preparation.

Competing interests. The authors have no competing interests to declare.

Funding. Work on Notch signalling in the Maillard laboratory is supported by the National Institute of Allergy and Infectious Diseases (grant no. R01-AI091627 to I.M.) and the Leukemia and Lymphoma Society (grant no. TRP 6583-20 to I.M.).

Acknowledgements. We thank Anneka Allman for assistance with figure design.

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
