## [Reviewer comments · Open Biology]

Review History

RSOB-19-0187.R0 (Original submission)

Review form: Reviewer 1

Recommendation

Accepts with minor revision (please list in comments)

Do you have any ethical concerns with this paper?

No

Comments to the Author

This review article is of great interest, since many open questions on the role of Notch Ligand/Receptor signaling in physiologic and tumor microenvironment are still unanswered as outlined by the Authors. Therefore there is the need to understand how Notch signaling mediates T cell crosstalk with stroma, and how stromal cells indulge on T cells with multiple ligands/chemokines signals. In that context I would suggest that Notch1 and Notch3 cooperate with many signaling pathways, including CXCR4 and Hedgehog. Indeed, I think that for a more comprehensive overview Notch3 should be included in figure 1, as well as discussed in the text, because this receptor is an integral component of the signaling and strictly related to T cell differentiation and function (in immature and mature T cells) and involved in T-cell leukemia development (Immunity 32, January 29, 2010; Biochem Biophys Res Commun. 2012 Feb

24;418(4):799-805; Oncogene. 2016 Nov 24;35(47):6077-6086; J Immunol Res. 2019 Jun 27;2019:5601396). Moreover, i would suggest, at lines 246-252 page 11, to discuss briefly the important role played by Notch and NF-kB in regulatory T cell responses (Front Immunol. 2018 Jun 4;9:1288.; Front Immunol. 2018 Oct 11;9:2165).

In figure 2, Interleukin-7 (IL7) as a target of Notch can be included in the table of Figure2.

Reference on page 10 line 220 seems to be not in line with the concept of the sentence.

Additionally, references should be added to the end of line 228 page 10 to sustain the concept.

Review form: Reviewer 2

Recommendation

Accept with minor revision (please list in comments)

Do you have any ethical concerns with this paper?

No

Comments to the Author

This review is really complete and well written. The authors did a really good job in summarizing the role of the Notch pathway in T cells.

Here are a couple of minor points that need to be addressed.

1) lines 187-206 - this part is a little bit too long and deviates a little bit from the main topic of the review.

2) given the key role of the Notch pathway in T cells differentiation, it is not surprising that some T cell malignancies (i.e. T-ALL) are characterized by Notch dysregulation. The paper would benefit from the presence of an additional (short) paragraph on this topic.

Decision letter (RSOB-19-0187.R0)

24-Sep-2019

Dear Dr Brandstadter

We are pleased to inform you that your manuscript RSOB-19-0187 entitled "Notch signaling in T cell homeostasis and differentiation" has been accepted by the Editor for publication in Open Biology. The reviewer(s) have recommended publication, but also suggest some minor revisions to your manuscript. Therefore, we invite you to respond to the reviewer(s)' comments and revise your manuscript.

Please submit the revised version of your manuscript within 7 days. If you do not think you will be able to meet this date please let us know immediately and we can extend this deadline for you.

- 1) A text file of the manuscript (doc, txt, rtf or tex), including the references, tables (including captions) and figure captions. Please remove any tracked changes from the text before submission. PDF files are not an accepted format for the "Main Document".
- 2) A separate electronic file of each figure (tiff, EPS or print-quality PDF preferred). The format should be produced directly from original creation package, or original software format. Please note that PowerPoint files are not accepted.
- 3) Electronic supplementary material: this should be contained in a separate file from the main text and meet our ESM criteria (see <http://royalsocietypublishing.org/instructions-authors#question5>). All supplementary materials accompanying an accepted article will be treated as in their final form. They will be published alongside the paper on the journal website and posted on the online figshare repository. Files on figshare will be made available approximately one week before the accompanying article so that the supplementary material can be attributed a unique DOI.

Online supplementary material will also carry the title and description provided during submission, so please ensure these are accurate and informative. Note that the Royal Society will not edit or typeset supplementary material and it will be hosted as provided. Please ensure that the supplementary material includes the paper details (authors, title, journal name, article DOI). Your article DOI will be 10.1098/rsob.2016[last 4 digits of e.g. 10.1098/rsob.20160049].

- 4) A media summary: a short non-technical summary (up to 100 words) of the key findings/importance of your manuscript. Please try to write in simple English, avoid jargon, explain the importance of the topic, outline the main implications and describe why this topic is newsworthy.

Images

Data-Sharing

It is a condition of publication that data supporting your paper are made available. Data should be made available either in the electronic supplementary material or through an appropriate repository. Details of how to access data should be included in your paper. Please see <http://royalsocietypublishing.org/site/authors/policy.xhtml#question6> for more details.

Data accessibility section

Sincerely,
The Open Biology Team
mailto:openbiology@royalsociety.org

Reviewer(s)' Comments to Author:

Referee: 1

Comments to the Author(s)

This review article is of great interest, since many open questions on the role of Notch Ligand/Receptor signaling in physiologic and tumor microenvironment are still unanswered as outlined by the Authors. Therefore there is the need to understand how Notch signaling mediates T cell crosstalk with stroma, and how stromal cells indulge on T cells with multiple ligands/chemokines signals. In that context I would suggest that Notch1 and Notch3 cooperate with many signaling pathways, including CXCR4 and Hedgehog. Indeed, I think that for a more comprehensive overview Notch3 should be included in figure 1, as well as discussed in the text, because this receptor is an integral component of the signaling and strictly related to T cell differentiation and function (in immature and mature T cells) and involved in T-cell leukemia development (Immunity 32, January 29, 2010; Biochem Biophys Res Commun. 2012 Feb 24;418(4):799-805; Oncogene. 2016 Nov 24;35(47):6077-6086; J Immunol Res. 2019 Jun 27;2019:5601396). Moreover, I would suggest, at lines 246-252 page 11, to discuss briefly the important role played by Notch and NF- κ B in regulatory T cell responses (Front Immunol. 2018 Jun 4;9:1288.; Front Immunol. 2018 Oct 11;9:2165).

In figure 2, Interleukin-7 (IL7) as a target of Notch can be included in the table of Figure2.

Reference on page 10 line 220 seems to be not in line with the concept of the sentence.

Additionally, references should be added to the end of line 228 page 10 to sustain the concept.

Referee: 2

Comments to the Author(s)

This review is really complete and well written. The authors did a really good job in summarizing the role of the Notch pathway in T cells.

Here are a couple of minor points that need to be addressed.

1) lines 187-206 - this part is a little bit too long and deviates a little bit from the main topic of the review.

2) given the key role of the Notch pathway in T cells differentiation, it is not surprising that some T cell malignancies (i.e. T-ALL) are characterized by Notch dysregulation. The paper would benefit from the presence of an additional (short) paragraph on this topic.

Author's Response to Decision Letter for (RSOB-19-0187.R0)

See Appendix A.

Decision letter (RSOB-19-0187.R1)

08-Oct-2019

Dear Dr Maillard

We are pleased to inform you that your manuscript entitled "Notch signaling in T cell homeostasis and differentiation" has been accepted by the Editor for publication in Open Biology.

Sincerely,

The Open Biology Team
mailto:openbiology@royalsociety.org

Appendix A

UNIVERSITY OF
PENNSYLVANIA
SCHOOL OF MEDICINE

Ivan Maillard, M.D., Ph.D.
Professor of Internal Medicine
Vice-Chief for Research, Division of Hematology-Oncology
Abramson Family Cancer Research Institute

Philadelphia, 10/05/2019

Re: Point-by-point response
Review Article, "Notch signaling in T cell homeostasis and differentiation"

Dear Editors:

We were delighted to hear about your acceptance of our submission for the review article entitled, "Notch signaling in T cell homeostasis and differentiation." We also appreciate the time that you and the referees have spent reviewing our manuscript and providing us with thoughtful feedback. Below is a summary of our response to concerns expressed by the reviewers, alongside a list of changes that we made to the manuscript to address these points.

Reviewer 1:

1) Indeed, I think that for a more comprehensive overview Notch3 should be included in figure 1, as well as discussed in the text, because this receptor is an integral component of the signaling and strictly related to T cell differentiation and function (in immature and mature T cells) and involved in T-cell leukemia development (Immunity 32, January 29, 2010; Biochem Biophys Res Commun. 2012 Feb 24;418(4):799-805; Oncogene. 2016 Nov 24;35(47):6077-6086; J Immunol Res. 2019 Jun 27;2019:5601396).

Our response: We have added additional discussion regarding the role of Notch3 in mature T cells (lines 216 - 221) and included references to a number of the papers recommended by the reviewer. This is in addition to our prior inclusion of a paper implicating Notch3 in autoimmunity (ref 65 in the revised manuscript), the only paper we are aware of to study Notch3 in mature T cells using the in vivo, loss-of-function approach that we emphasized throughout our review (other papers use gain-of-function strategies and/or are not focused on mature T cells).

2) Moreover, I would suggest, at lines 246-252 page 11, to discuss briefly the important role played by Notch and NF-κB in regulatory T cell responses (Front Immunol. 2018 Jun 4;9:1288.; Front Immunol. 2018 Oct 11;9:2165).

Our response: We already discuss the reported interplay between Notch and NF-κB signaling in the paper (lines 243 – 250). The two articles cited by the reviewer are both review articles including no references to primary research using the in vivo, loss-of-function approaches that we focused on throughout our review. We would favor not to include additional review articles in our list of references, as we think that our citations should be focused mostly on peer-reviewed primary data papers. Please let us know however if your editorial policies differ from this approach.

3) In figure 2, Interleukin-7 (IL7) as a target of Notch can be included in the table of Figure2. Reference on page 10 line 220 seems to be not in line with the concept of the sentence.

Our response: Thank you for this suggestion. We have added language specifically describing *IL7RA* as a proven Notch target in developing thymocytes and leukemic cells (lines 239 – 240). However, we are not aware of data indicating that this is also the case in mature T cells, the subject of this review. Indeed, transcriptional targets of Notch signaling can differ markedly in immature and mature T cells. Because Figure 2 was carefully constructed to focus on Notch targets in mature T cells, we would favor not to include *IL7RA* in this list, as evidence for its regulation by Notch in this context is lacking.

Additionally, references should be added to the end of line 228 page 10 to sustain the concept.

Our response: We have added a reference in this location, as recommended by the reviewer (line 242).

Reviewer 2:

1) lines 187-206 - this part is a little bit too long and deviates a little bit from the main topic of the review.

Our response: We have shorted the discussion on the role of Notch signaling in mature T cells in graft-versus-host disease per recommendation of the reviewer (lines 196 – 207), although we would like to point out that this is relevant to our central focus on mature T cells (rather than immature T cells or T cell leukemia).

2) given the key role of the Notch pathway in T cells differentiation, it is not surprising that some T cell malignancies (i.e. T-ALL) are characterized by Notch dysregulation. The paper would benefit from the presence of an additional (short) paragraph on this topic.

Our response: We have added a paragraph briefly reviewing Notch signaling in T-ALL as recommended by the reviewer (lines 86 – 93).

In summary, we are delighted that you find our review article to be suitable for publication in *Open Biology*. We have made significant additional effort to incorporate the feedback from our reviewers, while maintaining our paper's focus and systematic reference to rigorous supporting evidence.

Best regards,

Joshua Brandstadter, MD, PhD

Ivan Maillard, MD, PhD